# Screening Implications for Distribution of Colorectal Cancer Subsite by Age and Role of Flexible Sigmoidoscopy

**DOI:** 10.3390/cancers16061110

**Published:** 2024-03-10

**Authors:** Gloria Lin, David M. Hein, Po-Hong Liu, Amit G. Singal, Nina N. Sanford

**Affiliations:** 1Departments of Internal Medicine, The University of Texas Southwestern Medical Center, Dallas, TX 75390, USA; gloria.lin@utsouthwestern.edu (G.L.); po-hong.liu@utsouthwestern.edu (P.-H.L.); amit.singal@utsouthwestern.edu (A.G.S.); 2Departments of Radiation Oncology, The University of Texas Southwestern Medical Center, Dallas, TX 75390, USA; david.hein@utsouthwestern.edu

**Keywords:** colonoscopy, colorectal cancer, flexible sigmoidoscopy, early onset colorectal cancer

## Abstract

**Simple Summary:**

There is controversy over the optimal recommended screening practice for colorectal cancer. Between direct visualization methods, colonoscopy is more used more frequently, although randomized data suggest benefits from sigmoidoscopy rather than colonoscopy. We used the SEER database to assess the proportion of cancers that could be visualized with each screening strategy and found that 58% could be seen by sigmoidoscopy, including 73% of tumors for patients diagnosed younger than 50 years (early onset colorectal cancer). We recommend consideration of recommending flexible sigmoidoscopy as a population-based screening strategy for younger individuals who may be hesitant to undergo colonoscopy.

**Abstract:**

**Objectives:** The effectiveness of colonoscopy to reduce colorectal cancer (CRC) mortality is extrapolated from cohort studies in the absence of randomized controlled trial (RCT) data, whereas flexible sigmoidoscopy is supported by RCT data and may be easier to implement in practice. We characterized the anatomic distribution of CRC to determine the proportion that is visible with sigmoidoscopy. **Methods:** Patients with a primary diagnosis of colorectal adenocarcinoma were identified in the Surveillance, Epidemiology, and End Results program (2000–2020). Tumors from the rectum to the descending colon were categorized as visible by sigmoidoscopy, whereas more proximal tumors required colonoscopy. Differential prognosis between tumor locations, stratified by age groups and stage, was assessed using the overall restricted mean survival time (RMST) at 2, 5, and 10 years. **Results**: Among 309,466 patients, 58% had tumors visible by sigmoidoscopy, including 73% of those under age 50 (OR 2.10, 95% CI 2.03–2.16 age < 45, OR 2.20, 95% CI 2.13–2.27 age 45–49 versus age ≥ 50). Male sex (OR 1.54, 95% CI 1.51–1.56) and Asian or Pacific Islander race (OR 1.60, 95% CI 1.56–1.64) were also positively associated with tumors visualizable by sigmoidoscopy. Across age groups, for local disease, RMST was comparable for tumors visible versus not visible on sigmoidoscopy. For regional and metastatic cancer, patients with tumors visible by sigmoidoscopy had improved RMST versus those with more proximal tumors. **Conclusions**: 58% of CRC arises in locations visible by flexible sigmoidoscopy. Flexible sigmoidoscopy should be considered as a viable option for CRC screening, particularly in younger patients unwilling or unable to undergo colonoscopy.

## 1. Introduction

The goal of cancer screening is to detect malignancy at a treatable stage so that intervening based on the results of the test improves patient outcomes. In May 2021, the United States Preventive Services Task Force (USPSTF) lowered the age of screening initiation for average-risk individuals from 50 to 45 years [1]. Of note, those at greater risk may be recommended to start screening at an earlier age, and the task force recommends individuals discuss the advantages and disadvantages of screening and the type of testing with their physicians. Colonoscopy has long been the most prevalent form of screening modality in the United States, given its ability to visualize the entire colon and remove precancer lesions [2,3]. Accordingly, colonoscopies have been advocated by advocacy organizations and professional societies alike, with a recent emphasis on younger patients newly eligible for screening [4]. However, colonoscopy has potential downsides as compared to less invasive screening strategies: there is dependence on extensive bowel preparation for efficacy, the exam is conducted under sedation for most, and it is costly [5]. Additionally, there is a risk of severe adverse effects, including intestinal perforation and severe bleeding, at rates up to 9 and 36 per 10,000 procedures, respectively [6,7].

Until recently, two randomized trials demonstrated a reduction in death from CRC via screening from fecal occult blood tests (FOBT) and sigmoidoscopy [8,9,10]. In October 2022, the NordICC study, a trial of 80,000 patients randomized to invitation to colonoscopy screening versus none, found no reduction in death from CRC with screening invitation in the intention-to-treat analysis [11]. While colonoscopy reduced CRC mortality in per-protocol analysis, the low colonoscopy completion rate highlighted the challenges of colonoscopy-based screening programs.

Critical to the debate regarding screening strategies is the incidence and mortality of CRC by anatomic location. While flexible sigmoidoscopy is limited to examining the rectum, sigmoid colon, and descending colon, its benefits may be in adherence. To better compare screening options, understanding the proportion of cancers that can be seen by flexible sigmoidoscopy can better inform policy discussions on screening programs. As such, we used a large national cancer database to assess the proportion and prognosis of having a cancer in a location that could be visualized by a flexible sigmoidoscopy versus needing a complete colonoscopy, stratified by stage of diagnosis and patient age.

## 2. Methods

### 2.1. Patient Cohort

The Surveillance, Epidemiology, and End Results (SEER) program was used to identify individuals with cancers of the colon or rectum with adenocarcinoma histology (histologic type ICD-O-3 8140) diagnosed between 2000 and 2020 [12]. The SEER program used ICD-10 codes to determine primary CRC sites. Additional demographic and clinical variables collected included primary tumor site, age, sex, year of diagnosis, race and ethnicity, and stage (localized, regional, distant). Patients with unknown stages and those with the following non-specific tumor subsites were excluded: C18.8: Overlapping lesion of the colon, C18.9: Colon, NOS, and C26.0: Intestinal Tract, NOS. Patients diagnosed at autopsy and those with either incomplete survival data or 0 days of survival were also excluded. The year of diagnosis was grouped as follows: 2000–2004, 2005–2009, 2010–2014, and 2015–2020.

### 2.2. Cancer Site and Screening Modality

Tumors able to be visualized on sigmoidoscopy were those from the following primary subsites: C186: Descending Colon, C187: Sigmoid, C199: Rectosigmoid Junction, or C209: Rectum NOS. The remaining subsite locations were classified as those requiring full colonoscopy for detection and included: C180: Cecum, C182: Ascending Colon, C183: Hepatic Flexure, C184: Transverse Colon, or C185: Splenic Flexure.

### 2.3. Statistical Analysis

Our main outcome is the proportion of CRC tumors that could be visualized by sigmoidoscopy, stratified by age group and stage. Three age groups, <45 years, 45–49 years, and 50+ years, were defined based upon the recently lowered screening age eligibility of 45 years. We performed single and multivariable logistic regression to assess the odds of tumor visualization by sigmoidoscopy.

Overall survival (OS) from date of diagnosis was defined using the survival months and vital status recode variables. Proportional hazards assumptions were assessed with cloglog and Kaplan–Meier plots and analyses were stratified by age group and stage. Survival analysis was conducted using restricted mean survival time (RMST) and the difference in RMST (DRMST, RMST_visualizable_ − RMST_requires colonoscopy_) in each stage-age group strata at follow up times of 2, 5, and 10 years was reported. To reduce potential confounding in the RMST estimates, we first used inverse probability weighting (IPW) to produce Kaplan–Meier estimates of overall survival adjusted for sex, race, and ethnicity [13]. Standard errors were generated via bootstrapping (*n* = 2000), and the null hypothesis of no difference in RMST between groups was tested according to a Z-test [14,15]. To adjust for multiple testing of DRMST, a Bonferroni correction was performed based on the total number of tests performed (alpha = 0.05/27). We additionally compared the difference in RMST improvement of distant to localized disease by tumor location, stratified by age group (DDRMST = (RMST_visualizable localized_ − RMST_visualizable distant_) − (RMST_requires colonoscopy localized_ − RMST_requires colonoscopy distant_), alpha = 0.05/18). Analyses were conducted using R version 4.2.1 (R foundation, Vienna, Austria).

## 3. Results

A total of 309,466 patients had CRC identified in the SEER program. The median age was 66 years (IQR: 56–77), including 18,906 (6.1%) of patients aged 45–49 years and 5650 (4.4%) patients age < 45 years (Table 1). Across all age groups and stages, most tumors (58.2%) were in subsites able to be visualized by sigmoidoscopy (Figure 1, Table 1 and Table 2). This finding was most pronounced in younger patients. For example, for localized tumors, 71.0% vs. 54.9% of tumors were in subsites that could be assessed by sigmoidoscopy for patients ages 45–49 versus 50+ years old, respectively. For regional tumors, 74.2% vs. 55.2% of tumors were in subsites assessed by sigmoidoscopy for patients ages 45–49 versus 50+ years old, respectively. Overall, 73.1% of tumors were in locations able to be visualized by sigmoidoscopy in individuals younger than 50, as compared to 56.0% for those older than 50 years at diagnosis. The two most common tumor subsites were the rectum and sigmoid colon (22.3%, 22.2%, respectively).

Accordingly, on multivariable analysis, as compared to patients aged 50+, those <45 years and 45–49 years had 2.05 (95% CI 1.99–2.12) and 2.16 (95% CI 2.09–2.23) odds of having tumors that could be visualized on sigmoidoscopy, respectively (Table 1). Other variables positively associated with having tumors that could be visualized on sigmoidoscopy included male sex (OR 1.52, 95% CI 1.50–1.54) and Asian or Pacific Islander race (OR 1.56, 95% CI 1.52–1.60). In contrast, non-Hispanic Black patients had lower odds compared to non-Hispanic White patients for having tumors that could be seen on sigmoidoscopy (OR 0.76, 95% CI 0.75–0.78).

As expected, RMST at all time points and age groups was highest for localized disease and lowest for metastatic disease (Figure 2 and Figure 3, all RMST results found in Appendix A). For patients under 45 and over 50, tumors in locations requiring colonoscopy had greater differences in RMST between localized and distant stages when compared to tumors in locations seen by sigmoidoscopy. This difference was driven by worse prognoses for tumors requiring full colonoscopy at distant stages. For example, in patients under 45 at 120 months of follow-up, the difference in RMST between localized and distant disease for locations requiring colonoscopy was 82 months, while the difference for locations seen by sigmoidoscopy was 71 months (*p* < 0.001). This difference, although significant, was not as large for the 50+ age group, with RMST improvements of 64 vs. 60 months at 10 years of follow-up for the same comparison (*p* < 0.001).

In all age groups, for local disease, RMST was comparable for tumors in locations that could versus could not be visualized on sigmoidoscopy (Figure 2 and Figure 3, Appendix A). In contrast, for regional and metastatic cancer, RMST tended to be better for tumors that could be visualized by sigmoidoscopy across age groups. For example, RMST at 10 years for patients aged 50+ with regional disease was 74.9 versus 65.9 months for tumors that could not be seen on flexible sigmoidoscopy.

## 4. Discussion

In this study of patients diagnosed with CRC, we found that nearly two-thirds of tumors were in locations that could be visualized via a flexible sigmoidoscopy rather than requiring a full colonoscopy exam, and this difference was greatest in younger patients. These findings are timely, particularly in the context of the recently published NordICC trial, the updated USPSTF screening guidelines, and the increasing incidence of early-onset CRC [16].

In the NordICC trial, there was no difference in risk of colorectal cancer death or all-cause mortality between the groups that did or did not receive an invitation for colonoscopy screening. The study did find a modest difference in risk of developing colorectal cancer at 10 years, from 1.20 to 0.98%, with a number needed to screen of 455 to prevent 1 case of CRC. Reception of the study in the United States has been variable, with most relevant stakeholders emphasizing the actual screening rate of 42% in the study’s intervention group [17]. For proponents of colonoscopy, the argument is that a survival benefit to colonoscopy may be seen if a higher proportion of patients complete the exam. Furthermore, they emphasize that patients in the study received only a single invitation for screening, whereas primary care providers in the United States tend to have serial discussions of cancer screening with patients, which could increase uptake. Yet studies of colonoscopy screening at the population level in the United States have shown similar, less than optimal rates of compliance between 50% and 60% [18].

In comparison, the rate of screening completion in the largest randomized trial for flexible sigmoidoscopy was higher, at 63% [5]. While the reduction in CRC mortality could be due in part to this difference, our study suggests that another contributing factor could be that the majority of CRC can be visualized via a flexible sigmoidoscopy—that the additional 2/3 of the lower gastrointestinal tract seen on a complete colonoscopy does not translate to 66% of all CRC cases. In contrast, our study found that a full colonoscopy permitted visualization of approximately one-quarter of CRC in younger patients.

Focusing further on younger patients, the lowering of screening age was based upon modeling study projections and the rising incidence of colorectal cancer in individuals aged less than 50 years [19]. In modeling studies, “adherence” to screening is 100%, which is inconsistent with real-world practices. Furthermore, the rise in early-onset CRC has been largely driven by an increase in rectal cancer, and our study demonstrated that younger patients had the highest rate of having tumors that could be seen on a flexible sigmoidoscopy [20]. Among newly screen eligible patients aged 45–49 years, the rate of CRC screening is low, at 11.6% from January 2019 to August 2021 after guideline changes to include younger patients [21], and even this could be an overestimate because some of the tests could have been performed for diagnostic purposes in patients with symptoms.

Currently, the public health message targeting individuals in that age group is to seek CRC screening of any type, with colonoscopy being the gold standard. For young patients specifically, this is potentially problematic for several reasons. First, some could be hesitant to undergo a more invasive colonoscopy and would obtain no screening at all. While both colonoscopy and sigmoidoscopy are endoscopy-based, a colonoscopy is performed under deeper sedation and requires more extensive preparation. In addition, individuals could be pushed to undergo non-invasive testing. While some, such as FOBT and fecal immunohistochemical test (FIT), have data suggesting a benefit, others including novel blood-based methods, currently have no data showing a reduction in CRC incidence or mortality; these newer tests are also often costly and not covered by insurance [22]. Third, studies have shown racial/ethnic disparities in screening uptake that are more pronounced in younger versus older patients. This may be due in part to cultural differences in acceptance of invasive screening procedures [2]. For example, one study found that non-white individuals had the highest adherence to FOBT, while white participants adhered more often to colonoscopy [18]. As such, a screening campaign focusing on colonoscopy could further exacerbate disparities in screening rates. Given the above concerns and the fact that most tumors in this population can be detected on a less invasive flexible sigmoidoscopy, organizations should consider recommending flexible sigmoidoscopy as a pragmatic screening method for younger individuals newly eligible for CRC screening. In addition, with epidemiologic trends of increasing CRC among patients in the 40–44 age group with an even greater predominance of distal cancers, it would be reasonable to consider assessing the benefits of a flexible sigmoidoscopy in this younger population currently not eligible for screening [16].

We showed that RMST for non-metastatic tumors was better than for metastatic tumors, with a greater difference in RMST seen in tumor locations requiring colonoscopy. Furthermore, cancers that could be visualized on sigmoidoscopy had a slightly better prognosis at regional and distant stages and across age groups as compared to those requiring a full colonoscopy. Since the goal of cancer screening is to detect disease prior to the development of metastases, these findings underscore the need to balance advantages and disadvantages of different screening strategies. The advantages of sigmoidoscopy likely include better compliance, which can then translate into improved effectiveness. Colonoscopies could detect cancers with worse prognoses earlier, but this may not outweigh the aggregate lower adherence rates compared to sigmoidoscopy or other screening methods. While screening compliance in the NordICC study with invitation to colonoscopy was 42%, adherence to flexible sigmoidoscopy was reportedly higher in prior randomized controlled trials, ranging from 58 to 83% [23]. The role of sigmoidoscopy as an acceptable alternative for colonoscopy in patient preference continues to be further explored, and our study contributes to characterizing the utility of sigmoidoscopy in a more personalized screening process [24].

Limitations of our study include a lack of information on familial disposition or other risk factors for CRC, such as inherited syndromes or inflammatory bowel disease. In addition, there were no details of treatment; prior studies have shown that younger patients receive more intensive treatment for similar cancer stages [25]. As such, our results are separated by age groups who should have more similar performance status and treatment tolerability. In addition, we assessed tumor location and detectability based upon two screening methods: flexible sigmoidoscopy and colonoscopy, and we did not discuss other modalities. There are currently two ongoing randomized studies comparing FOBT versus colonoscopy and annual FIT versus colonoscopy that will further assess the benefit of colonoscopy versus other screening strategies that are non-invasive [26]. Studies assessing combination modalities (for example, sigmoidoscopy + FOBT versus colonoscopy) would also be of interest.

## 5. Conclusions

In summary, most CRCs arise in locations detectable by flexible sigmoidoscopy, particularly in younger patients. Screening with sigmoidoscopy is better than no testing at all and can detect the majority of cancers, although some argue that colonoscopy could be the gold standard if compliance were higher. However, our findings, in the context of the recently negative NordICC trial, call into question the advantages and disadvantages of a “colonoscopy first” public health recommendation for CRC screening in healthy asymptomatic individuals and could support consideration of flexible sigmoidoscopy as an additional initial option for CRC screening.

## Figures and Tables

**Figure 1 cancers-16-01110-f001:**
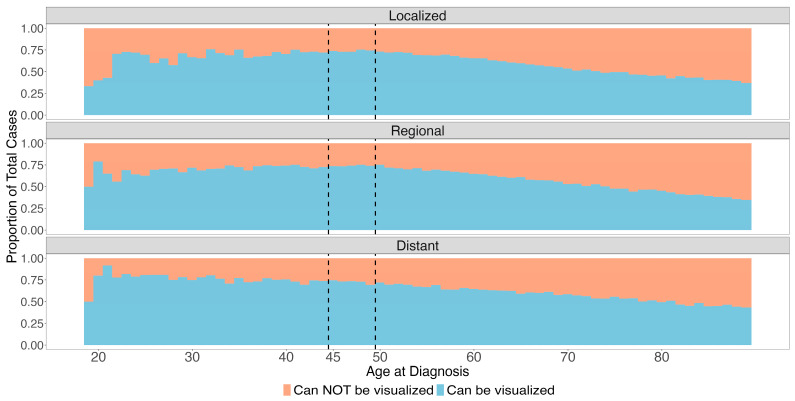
Proportion of tumors in sites able to be visualized by sigmoidoscopy by age and stage.

**Figure 2 cancers-16-01110-f002:**
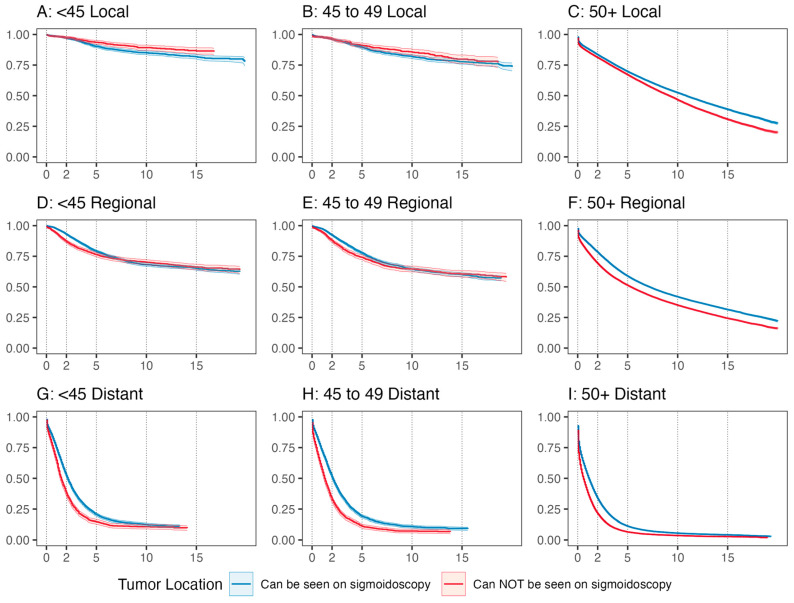
Kaplan–Meier survival curves by age group, stage, and tumor location. Curves have been adjusted for sex and race and ethnicity using inverse probability weights.

**Figure 3 cancers-16-01110-f003:**
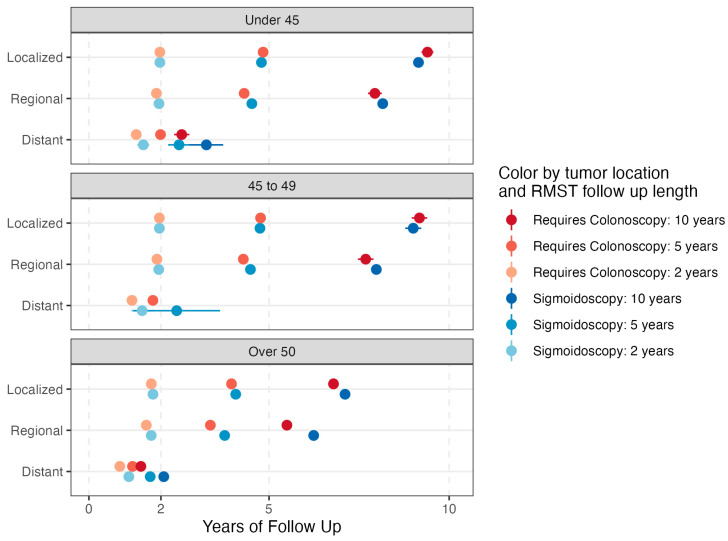
RMST at 2, 5, and 10 years of follow-up for tumor locations requiring colonoscopy vs. able to be visualized on sigmoidoscopy, stratified by age group and stage and adjusted for sex, race, and ethnicity. Notably, RMST for the 45–49 age group with distant disease at 120 months could not be assessed due to insufficient sample size. Bootstrapped (2000 replicates) 99% confidence intervals included.

**Table 1 cancers-16-01110-t001:** Cohort characteristics and odds of tumor location able to be visualized by sigmoidoscopy.

	N (%) Can Be Visualized	N Total	Univariable OR (95% CI)	Multivariable OR (95% CI)
**Overall**	179,986 (58.2)	309,466	-	-
**Age Group**				
Over 50	151,015 (56.0)	269,861	ref	ref
45–49	13,922 (73.6)	18,906	2.20 (2.13, 2.27)	2.16 (2.09, 2.23)
Under 45	15,049 (72.7)	20,699	2.10 (2.03, 2.16)	2.05 (1.99, 2.12)
**Sex**				
Female	78,119 (52.7)	148,136	ref	ref
Male	101,867 (63.1)	161,330	1.54 (1.51–1.56)	1.52 (1.50, 1.54)
**Cancer Stage**				
Localized	54,769 (56.6)	96,974	ref	ref
Regional	77,596 (57.6)	134,694	1.05 (1.03, 1.06)	1.01 ^†^ (0.99, 1.03)
Distant	47,621 (61.2)	77,798	1.22 (1.19, 1.24)	1.16 (1.14, 1.19)
**Race**				
Non-Hispanic White	117,647 (57.1)	206,185	ref	ref
Hispanic (All Races)	23,089 (62.9)	36,736	1.27 (1.24, 1.30)	1.18 (1.15, 1.21)
Non-Hispanic American Indian/Alaska Native	1481 (61.9)	2394	1.22 (1.12, 1.33)	1.17 (1.07, 1.27)
Non-Hispanic Asian or Pacific Islander	19,070 (68.0)	28,041	1.60 (1.56, 1.64)	1.56 (1.52, 1.60)
Non-Hispanic Black	18,018 (51.4)	35,067	0.80 (0.78, 0.81)	0.76 (0.75, 0.78)
Non-Hispanic Unknown Race	681 (65.3)	1043	1.42 (1.25, 1.61)	1.35 (1.19, 1.54)
**Year of Diagnosis**				
2000–2004	38,201 (57.2)	66,774	ref	ref
2005–2009	39,821 (57.2)	69,575	1.00 (0.98, 1.02)	0.98 (0.96, 1.00)
2010–2014	40,625 (58.0)	70,071	1.03 (1.01, 1.05)	0.99 (0.97. 1.01)
2015–2020	61,339 (60.0)	103,046	1.10 (1.08, 1.12)	1.03 (1.01, 1.05)

^†^: Indicates not-significant at alpha = 0.05.

**Table 2 cancers-16-01110-t002:** Tumor sites by age group.

Primary Tumor Site	Age Group			
	<45	45–49	>50	All Ages
**Rectum**	6152	5807	56,988	68,947 (22.3%)
**Rectosigmoid Junction**	2333	2186	23,762	28,281 (9.14%)
**Sigmoid**	5380	4976	58,308	68,664 (22.2%)
**Descending Colon**	1184	953	11,957	14,094 (4.55%)
**Splenic Flexure**	571	443	6922	7936 (2.56%)
**Transverse Colon**	1146	878	18,595	20,619 (6.66%)
**Hepatic Flexure**	563	418	9887	10,868 (3.51%)
**Ascending Colon**	1615	1493	38,933	42,041 (13.6%)
**Cecum**	1755	1752	44,509	48,016 (15.5%)
**All Sites**	20,699 (6.69%)	18,906 (6.11%)	269,861 (87.2%)	309,466 (100%)

## Data Availability

R code available at https://github.com/DavidHein96/seer-sigmoidoscopy.

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
