# Peer review of "Screening Implications for Distribution of Colorectal Cancer Subsite by Age and Role of Flexible Sigmoidoscopy"

_cancers, 2024, doi:10.3390/cancers16061110_

Round 1
Reviewer 1 Report
Comments and Suggestions for Authors
-
The authors provide an interesting SEER analysis looking at the number of newly diagnosed colorectal cancers that would be detectable by sigmoidoscopy versus colonoscopy (as defined by anatomic location). This manuscript provides important nuance to the colon cancer screening discussion, and the tradeoffs involved, particularly in light of the recently published NordICC study. The authors should be commended for the clarity of writing and excellent presentation of the analysis and data. I have only a couple comments for the authors to consider that may strengthen their manuscript:
-
Major comment:
-
I worry the authors come close to overstating their findings. For example in the Abstract Conclusion to say that “most” CRC arises in locations visible by flexible sigmoidoscopy may be technically true (58%), but this ignore the other side of the coin, which is that sigmoidoscopy still would have missed 1-58%=42% of CRC cancers by the authors definition. “Most” is overly reductive and at risk of being misleading. My recommendation would be for the authors to soften their language in the Abstract and Discussion/Conclusions and adopt a position more of “screening with sigmoidoscopy is better than no screening, and does a good job at screening catching the majority of cancers (58%), particularly in younger patients, but the gold standard, assuming compliance, is colonoscopy”
-
Minor Comments (do not have to be addressed):
-
It would have been interesting in the Discussion if the authors had commented on the role of sigmoidoscopy + FOBT. Put another way, I wonder how much the sensitivity of screening improved by combining ‘less invasive’ methods. Another way to look at it: What about patients that have a positive FOBT but a negative sigmoidoscopy (how much does FOBT close the gap between colonoscopy and sigmoidoscopy; I honestly dont know the answer).
-
The acronym FIT is used without being defined (fecal immunochemical test)
Author Response
The authors provide an interesting SEER analysis looking at the number of newly diagnosed colorectal cancers that would be detectable by sigmoidoscopy versus colonoscopy (as defined by anatomic location). This manuscript provides important nuance to the colon cancer screening discussion, and the tradeoffs involved, particularly in light of the recently published NordICC study. The authors should be commended for the clarity of writing and excellent presentation of the analysis and data. I have only a couple comments for the authors to consider that may strengthen their manuscript:
- Thank you so much for your kind review of our work.
Major comment:
I worry the authors come close to overstating their findings. For example in the Abstract Conclusion to say that “most” CRC arises in locations visible by flexible sigmoidoscopy may be technically true (58%), but this ignore the other side of the coin, which is that sigmoidoscopy still would have missed 1-58%=42% of CRC cancers by the authors definition. “Most” is overly reductive and at risk of being misleading. My recommendation would be for the authors to soften their language in the Abstract and Discussion/Conclusions and adopt a position more of “screening with sigmoidoscopy is better than no screening, and does a good job at screening catching the majority of cancers (58%), particularly in younger patients, but the gold standard, assuming compliance, is colonoscopy”
- Thank you very much for taking the time to review our paper. We have modified our language accordingly.
- Changed in text: 58% of CRC arise in locations visible by flexible sigmoidoscopy. Flexible sigmoidoscopy should be considered as a viable option for CRC screening, particularly in younger patients unwilling or unable to undergo colonoscopy.
- Added in text: In summary, most CRC arise in locations detectable by flexible sigmoidoscopy, particularly in younger patients. Screening with sigmoidoscopy is better than no testing at all and can detect the majority of cancers, although some argue that colonoscopy could be the gold standard if compliance were higher.
Minor Comments (do not have to be addressed):
It would have been interesting in the Discussion if the authors had commented on the role of sigmoidoscopy + FOBT. Put another way, I wonder how much the sensitivity of screening improved by combining ‘less invasive’ methods. Another way to look at it: What about patients that have a positive FOBT but a negative sigmoidoscopy (how much does FOBT close the gap between colonoscopy and sigmoidoscopy; I honestly dont know the answer).
- Great idea. Added to discussion.
- Added in text: Studies assessing combination modalities (for example, sigmoidoscopy + FOBT versus colonoscopy) would also be of interest.
The acronym FIT is used without being defined (fecal immunochemical test)
- Response: Thank you for catching this, we have defined the acronym.
Reviewer 2 Report
Comments and Suggestions for Authors
Thank you for the opportunity to review this manuscript.
Overall the manuscript is well written. However I suggest a few changes to improve this manuscript.
In particular, in the introduction I think that it necessary to better clarify when to start screening and the reasons that are related. The U.S. Preventive Services Task Force (Task Force) recommends that adults age 45 to 75 be screened for colorectal cancer but People at an increased risk of getting colorectal cancer should talk to their doctor about when to begin screening, which test is right for them, and how often to get tested.
Methods and results are well written.
In the discussion, in the last paragraph, concludes with "currently, the public health message targeting individuals in that age group is to 190 seek CRC screening of any type, with colonoscopy being the gold", I suggest to write in comparison, what this study suggests.
Author Response
Thank you for the opportunity to review this manuscript.
Overall the manuscript is well written. However I suggest a few changes to improve this manuscript.
- Response: Thank you for taking the time to review our work, for the kind assessment, and for the suggestions for edits.
In particular, in the introduction I think that it necessary to better clarify when to start screening and the reasons that are related. The U.S. Preventive Services Task Force (Task Force) recommends that adults age 45 to 75 be screened for colorectal cancer but People at an increased risk of getting colorectal cancer should talk to their doctor about when to begin screening, which test is right for them, and how often to get tested.
- Response: Thank you. We have added this nuance to the introduction.
- Added in text: Of note, those at greater risk may be recommended to start screening at an earlier age, and the task force recommends individuals discuss the pros and cons of screening and type of testing with their physicians.
Methods and results are well written.
- Response: Thank you.
In the discussion, in the last paragraph, concludes with "currently, the public health message targeting individuals in that age group is to 190 seek CRC screening of any type, with colonoscopy being the gold", I suggest to write in comparison, what this study suggests.
- Response: We have added this to the last paragraph of our manuscript.
- Added to text: Screening with sigmoidoscopy is better than no testing at all and can detect the majority of cancers, although some argue that colonoscopy could be the gold standard if compliance were higher. However, our findings, in the context of the recently negative NordICC trial, call into question the pros and cons of a “colonoscopy first” public health recommendation for CRC screening in healthy asymptomatic individuals, and could support consideration of flexible sigmoidoscopy as an additional initial option for CRC screening.